# Persistent effects of pair bonding in lung cancer cell growth in monogamous *Peromyscus californicus*

Asieh Naderi[1], Elham Soltanmaohammadi[1], Vimala Kaza[2], Shayne Barlow[3], Ioulia Chatzistamou[4], Hippokratis Kiaris[1,2]*

[1]Department of Drug Discovery and Biomedical Sciences, College of Pharmacy, University of South Carolina, Columbia, United States; [2]Peromyscus Genetic Stock Center, University of South Carolina, Columbia, United States; [3]Department of Cell Biology and Anatomy, School of Medicine, and Animal Resources Facility, University of South Carolina, Columbia, United States; [4]Department of Pathology, Microbiology and Immunology, School of Medicine, University of South Carolina, Columbia, United States

**Abstract** Epidemiological evidence suggests that social interactions and especially bonding between couples influence tumorigenesis, yet whether this is due to lifestyle changes, homogamy (likelihood of individuals to marry people of similar health), or directly associated with host-induced effects in tumors remains debatable. In the present study, we explored if tumorigenesis is associated with the bonding experience in monogamous rodents at which disruption of pair bonds is linked to anxiety and stress. Comparison of lung cancer cell spheroids that formed in the presence of sera from bonded and bond-disrupted deer mice showed that in monogamous *Peromyscus polionotus* and *Peromyscus californicus*, but not in polygamous *Peromyscus maniculatus*, the disruption of pair bonds altered the size and morphology of spheroids in a manner that is consistent with the acquisition of increased oncogenic potential. In vivo, consecutive transplantation of human lung cancer cells between *P. californicus*, differing in bonding experiences (n = 9 for bonded and n = 7 for bond-disrupted), and nude mice showed that bonding suppressed tumorigenicity in nude mice (p<0.05), suggesting that the protective effects of pair bonds persisted even after bonding ceased. Unsupervised hierarchical clustering indicated that the transcriptomes of lung cancer cells clustered according to the serum donors' bonding history while differential gene expression analysis pointed to changes in cell adhesion and migration. The results highlight the pro-oncogenic effects of pair-bond disruption, point to the acquisition of expression signatures in cancer cells that are relevant to the bonding experiences of serum donors, and question the ability of conventional mouse models to capture the whole spectrum of the impact of the host in tumorigenesis.

*For correspondence: kiarish@cop.sc.edu

Competing interests: The authors declare that no competing interests exist.

## Introduction

While the psychosomatic impact of cancer in patients is extensively documented, the reciprocal effects of individuals' social experiences in carcinogenesis receive limited attention. Both anecdotal and experiential evidence, and numerous epidemiological studies, strongly suggest that emotional factors can affect the development and progression of cancer, pointing to the sensitivity of cancer cells to signals associated with behavior, emotional state, and sociality. For example, the marital status modulates the likelihood for the development of fatal cancers, with unmarried, divorced, or widowed individuals exhibiting an increased chance of developing life-threatening disease and males being more susceptible than females to the protective effects of marriage (*Aizer et al., 2013*). The

**eLife digest :** People's social interactions could influence their risk of developing various diseases, including cancer, according to population-level studies. In particular, studies have identified a so-called widowhood effect where a person's risk of disease increases following the loss of a spouse. However, the cause of the widowhood effect remains debatable, as it can be difficult to separate the impact of lifestyle changes from biological changes in the individual following bereavement.

It is not possible to use laboratory mice to identify a causal biological mechanism, because they do not form long-term relationships with a single partner (pair bonds). However, several species of deer mouse form pair bonds, and suffer from anxiety and stress if these bonds are broken. Naderi et al. used these mice to study the widowhood effect on the risk of developing cancer.

First, Naderi et al. grew human lung cancer cells in blood serum taken from mice that were either in a pair bond or had been separated from their partner. The cancer cells grown in the blood of mice with disrupted pair bonds changed size and shape, indicating that these mice were more likely to develop cancer. This effect was not observed when the cells were grown in the blood of bonded deer mice or of another deer mouse species that does not form pair bonds. Naderi et al. also found that the activity of genes involved in the cancer cells' ability to spread and to stick together was different in pair-bonded mice and in pair-separated mice.

Next, Naderi et al. implanted lung cancer cells into the deer mice to study their effects on live animals. When cancer cells from the deer mice were transplanted into laboratory mice with a weakened immune system, the cells taken from pair-bonded deer mice were less likely to grow than the cells from deer mice with disrupted pair bonds. This suggests that the protective effects of pair bonding persist even after removal from the original mouse.

These results provide evidence for a biological mechanism of the widowhood effect, where social experiences can alter gene activity relating to cancer growth. In the future, it will be important to determine whether the same applies to humans, and to find out if there are ways to mimic the effects of long-term bonds to improve cancer prognoses.

'widowhood effect' provides an example at which in couples, after the loss of one partner, the surviving one exhibits an increased probability for the development of various fatal pathologies (*Elwert and Christakis, 2008*; *Blanner et al., 2020*; *Sullivan and Fenelon, 2014*; *Bowling, 1987*; *Boyle et al., 2011*). Notwithstanding that high variation in death causes has been documented, cancer is recognized as a common cause of mortality (*Aizer et al., 2013*; *Elwert and Christakis, 2008*; *Blanner et al., 2020*; *Burgoa, 1998*; *Martikainen and Valkonen, 1996*; *Sex, 1973*). Although both sexes are influenced by widowhood, males appear more sensitive than females to widowhood-associated death (*Sullivan and Fenelon, 2014*; *Helsing et al., 1981*).

Despite the information they provide, unavoidable changes in lifestyle habits in the bereaved partner at widowhood or between single and married patients complicate the epidemiological data interpretation. Several mechanisms connecting cancer to social interactions, mental state, and bereavement have been proposed. Laboratory mice of the genus Mus, despite their power in illuminating various aspects of tumorigenesis, remain of limited value in modeling the effects of pair bonding. It is estimated that in less than 10% of mammals, including humans, individuals form pair bonds that are based on mating (*Kleiman, 1977*; *Lukas and Clutton-Brock, 2013*; *Scribner et al., 2020*). Therefore, mice, by not developing long-term pair bonds, are not adequate in studying the effects of widowhood and pair-bond disruption (*Chatzistamou et al., 2018*; *McDonald et al., 2005*). Earlier studies in mice have shown that brain-derived signals linked to the reward system may impact tumorigenesis, whereas stress can stimulate metastases (*Ben-Shaanan et al., 2018*; *Sloan et al., 2010*). However, more complex behavioral traits involving social interactions in married couples or widowhood cannot be studied in mice. *Peromyscus californicus* is a monogamous species developing long-term, cohesive pair bonds that can influence various physiological responses (*Havighorst et al., 2017*; *Perea-Rodriguez et al., 2015*; *Wright et al., 2018*). Upon cyclosporine-mediated immunosuppression, similarly with other rodents, *P. californicus* supports the growth of human cancers, providing a potentially informative animal model for the study

of pair-bond disruption in tumorigenesis in vivo (*Glasper and Devries, 2005*; *Fingert et al., 1984*; *Kaza et al., 2018*; *Chatzistamou and Kiaris, 2016*).

## Results

### Bonding history modulates the effects of sera in tumor spheroid formation

Initially, we asked if sera of *P. californicus* following the disruption of pair bonds affected the growth of cancer cells in vitro in a manner that depended on bonding history. We focused on the formation of tumor spheroids that are enriched in cells with cancer stem cell (CSC)-like properties, and their formation is known to reflect tumorigenic activity directly (*Visvader and Lindeman, 2012*; *Ishiguro et al., 2017*). Sera were obtained from 14 to 17 months old virgin, bonded for about 12 months, or bond-disrupted (after 12 months of bonding) at the periods indicated, male *P. californicus*, and the efficacy of spheroid formation by A549 human lung cancer cells was assessed. A pilot study indicated that sera obtained from animals 1 week after the disruption of pair bonds resulted in the formation of larger yet less compact spheroids, suggesting a significant impact of bond disruption in spheroid morphogenesis (*Figure 1a*). The results were confirmed and extended in a subsequent study that also included sera obtained 24 hr and 2 weeks after the disruption of pair bonds (*Figure 1b*). In this study, sera from 9 (B), 5 (BD, 24 hr), 5 (BD, 1 week), 4 (BD, 2 weeks), and 5 (virgin, V) different animals were used, and microsphere formation was evaluated in two biological replicas for each (n = 10 for BD [1 week], BD [24 hr], and V; n = 8 for BD [2 weeks] and n = 18 for B). For control media (CM) and plain serum-free media (PM), n = 4. As shown in *Figure 1b*, this activity was only marginal at 24 hr but was significant ($p < 0.05$) 1 week and 2 weeks after the disruption of pair bonds, implying that the factors responsible accumulated in the sera after pair-bond disruption. As compared to virgins, sera from animals at bonding resulted in the formation of smaller spheroids, albeit insignificantly, which implies that bonding may also have some protective activity, beyond the pro-oncogenic activity of bond disruption (*Figure 1b*).

### Variation in spheroid size with sera obtained after bond disruption is due to the genetic diversity of donor animals and persists in different lung cancer cell lines

 The effects in spheroid size described above were obtained with sera from older animals (14–17 months old) that were bonded for at least 12 months. To test whether disruption of bonds in younger animals that were bonded for shorter time periods also produced similar effects, we conducted the following study: We exposed to sera of 8–10 months old animals that were either pair-bonded for 2 months or following 2 weeks of bond disruption after 2 months of bonding, a roster of lung cancer cell lines. For this experiment, 14 animals were used that represented seven sibling pairs with each sibling allocated either to the bonded or to the bond-disrupted group. Our results indicated that consistently, in the same sibling pair, an induction of microsphere size of similar magnitude was noted for all five cell lines tested, in four of seven pairs, while this effect was only marginal in the remaining three pairs (*Figure 1—figure supplement 1*). The variation in spheroid size was analogous to that recorded in the results described in *Figure 1c*. Thus, we conclude that even shorter periods of bonding are sufficient, and the consequences of its disruption can be recorded in sera from even younger animals. More importantly, it indicates that the variation of the effects is due to the diversity of the animals and not to the differential sensitivity of the cell lines used.

### Persistent pro-oncogenic activity of bond disruption in vivo

The effects of pair bonding in spheroid formation prompted us to explore whether bond disruption also influences the efficacy of tumorigenesis in vivo. To that end, vasectomized male *P. californicus* were allowed to establish pair bonds for about 2 months with their female partners and then subjected to pair-bond disruption (n = 9) or were left with their partners (n = 11). Following immunosuppression by CsA animals were inoculated with A549 human lung cancer cells and tumorigenesis was monitored. Animals that did not possess bonding experiences before were used as controls (n = 8). Tumors grew originally in animals of all experimental groups and by day 15 measurable tumors were detected in 9 of 11 bonded, in 8 of 9 bond-disrupted and in 6 of 8 virgins (*Figure 2a*). At this point,

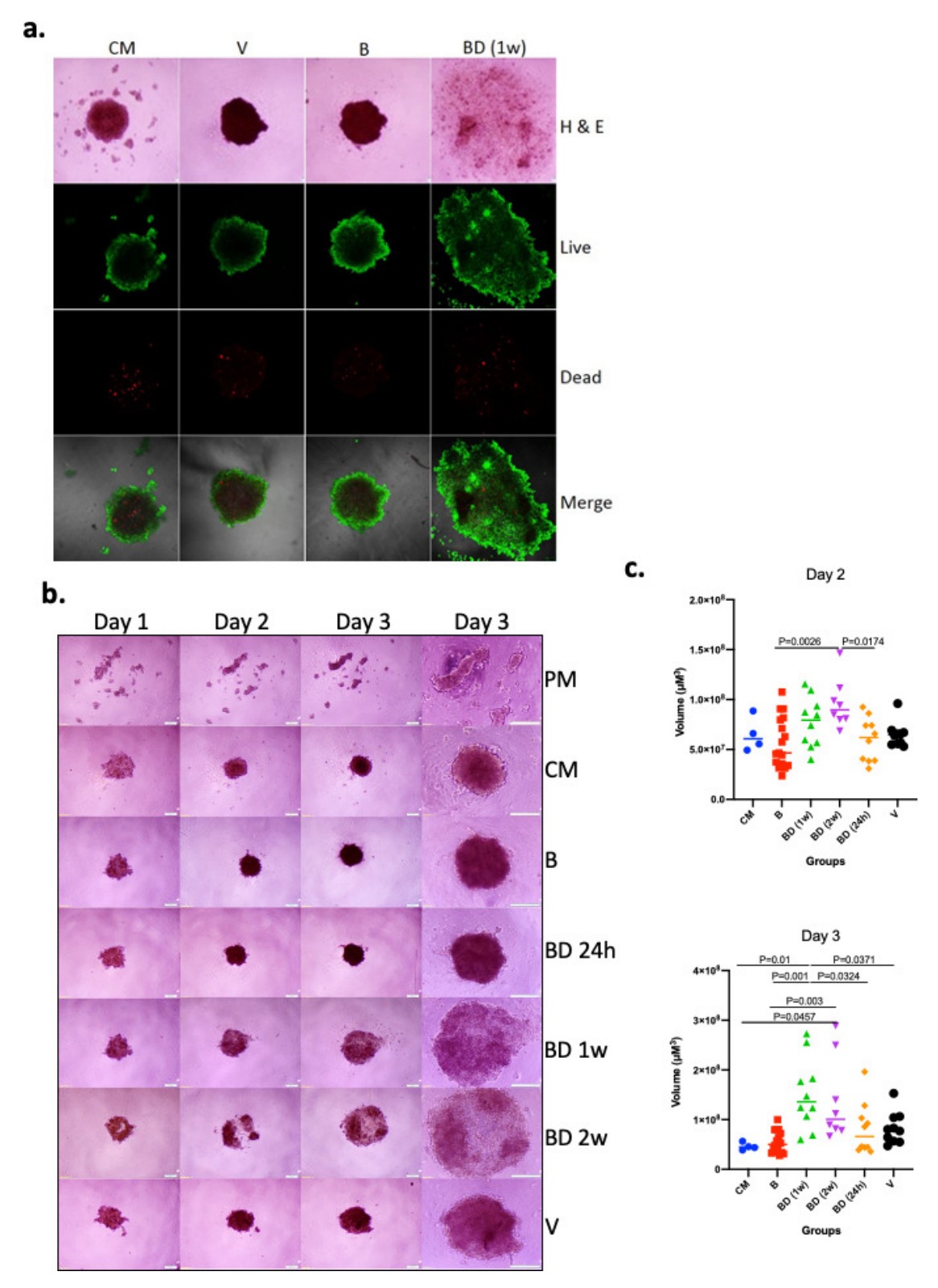

**Figure 1.** Effects of pair bonding in the pro-oncogenic activity of sera. (a) Representative microphotographs of tumor spheroids developed by A549 cells 3 days after cell seeding. Cells formed spheroids in the presence of sera from bonded (B), bond-disrupted animals (BD) 1 week after disruption, and virgin animals (V), and control media containing fetal bovine serum (FBS) (CM). Live (green) and dead cells (red) are indicated. Representative images of two independent experiments are shown. (b) Representative microphotographs of tumor spheroids developed by A549 cells on day 1, day 2,

*Figure 1 continued on next page*

*Figure 1 continued*

and day 3, after cell seeding. Cells formed spheroids in the presence of sera from bonded (B), bond-disrupted animals (BD) 24 hr, 1 week, and 2 weeks after disruption, and virgin animals (V), control media containing FBS (CM) or serum-free plain media (PM). The last column shows images at day three in higher magnification. Bars indicate 200 μM. (c) Scatter dot plots of data shown in (b), indicating the size of tumor spheroids at days 2 and 3 after seeding. Median and p-values are indicated. Sera from 9 (B), 5 (BD, 24 hr), 5 (BD, 1 week), 4 (BD, 2 weeks), and 5 (V) different animals were used and microsphere formation was evaluated in two biological replicas for each (n = 10 for BD [1 week], BD [24 hr] and V; n = 8 for BD [2 weeks] and n = 18 for B). For control media (CM) and plain serum-free media (PM), n = 4. For this experiment, sera from 14 to 17 months old mice that were used for the B and BD groups were bonded for about 1 year. Statistical analyses were performed by ANOVA.

The online version of this article includes the following figure supplement(s) for figure 1:

**Figure supplement 1.** Microsphere morphology of a panel of human lung cancer cells cultured in *P. californicus* sera.

tumors were modestly – albeit not statistically significantly – larger in the bond-disrupted animals and smaller in the group of virgins (*Figure 2a*). By day 25, the tumors persisted in both the bond-disrupted and bonded animals, at 89% (8 of 9) and 82% (9 of 11) rate, respectively, while in virgin animals, they were detectable only in 25% (2 of 8) of the animals (*Figure 2b,c*).

In a follow-up study, we explored if differential pro-oncogenic activity persisted after growth in nude mice. Thus, tumors that were originally grown in *P. californicus* for at least 1 month (n = 9 for bonded and n = 7 for bond-disrupted) were re-transplanted in virgin nude mice (one nude mice for each original Peromyscus tumor), and tumorigenesis was recorded. As shown in *Figure 2d*, tumors from bonded *P. californicus* exhibited significantly (p=0.011) lower tumorigenicity in nude mice than those grown originally in the bond-disrupted animals, despite that histologically they remained indistinguishable (*Figure 2e*). In line with the tumor spheroid analyses, pair bonding produced persistent changes in tumors that suppressed their growth and endured even when bonding seized.

## Effects of pair bonding in differential gene expression

The effects of bonding history in the profile of tumor growth in vivo, combined with the spheroid formation in vitro, imply the induction of transcriptional changes in the cancer cells in a manner that depends on bonding experience (*Figure 3*, *Figure 3—figure supplements 1*, *2* and *3*). Initially, we focused on the expression of established CSC markers and genes regulating CSC potential, such as Oct-4, b-catenin, and CD-133 that have been identified previously in A549 cells (*Chiou et al., 2010*; *Akunuru et al., 2012*; *Teng et al., 2010*). The analysis was performed by semiquantitative RT-PCR in 2D cultures to eliminate the effects of the clonal selection of cells in the spheroids. Differential expression analysis did not reveal considerable differences between the bonding groups, either in cells cultured in vitro with sera from animals differing in bonding history or in vivo in tumors in nude mice or Peromyscus (*Figure 3—figure supplement 1*). However, unsupervised hierarchical clustering indicated that these CSC markers provided a signature that predicted a relatively high accuracy bonding history of the animals (*Figure 3—figure supplement 1*).

This observation prompted us to perform RNA sequencing and analyze expression profiles at the whole transcriptome level in human A549 lung cancer cells in the presence of sera that had been isolated from monogamous male *P. californicus* that were virgin (V), bonded (B), or subjected to disruption of pair bonds (BD) after bonding (n = 6 samples/group). Controls (C) cultured in the presence of fetal bovine serum (FBS) were also included. Unsupervised hierarchical clustering (*Vidman et al., 2019*) indicated that the transcriptomes clustered well together according to the serum donors' bonding history, except the virgin (V) group that exhibited the lowest discrimination (*Figure 3—figure supplement 2*). Differential gene expression analysis was performed as described before by using the iDEP platform (*Ge et al., 2018*). This analysis showed that the majority of differentially expressed genes were detected in the comparisons involving the FBS-treated cells (C), which suggests that the species origin of sera produces the most potent effects in gene expression and potentially masking the consequences of pair bonding in the regulation of the transcriptome (*Figure 3—figure supplement 3*). Thus, we repeated the analysis by excluding the specimens corresponding to FBS and restricted it only to the specimens that received Peromyscus sera (*Figure 3*). Seven genes were differentially expressed in each B vs BD and B vs V comparisons, while none were detected between the V and BD groups (*Table 1*). Thus, it seems that pair bonding produces more robust effects in the sera as compared to those of bond disruption. Among these genes, all of which were downregulated in the B group, five were common and included HES1, ZFP36, NR4A1, FGG, and

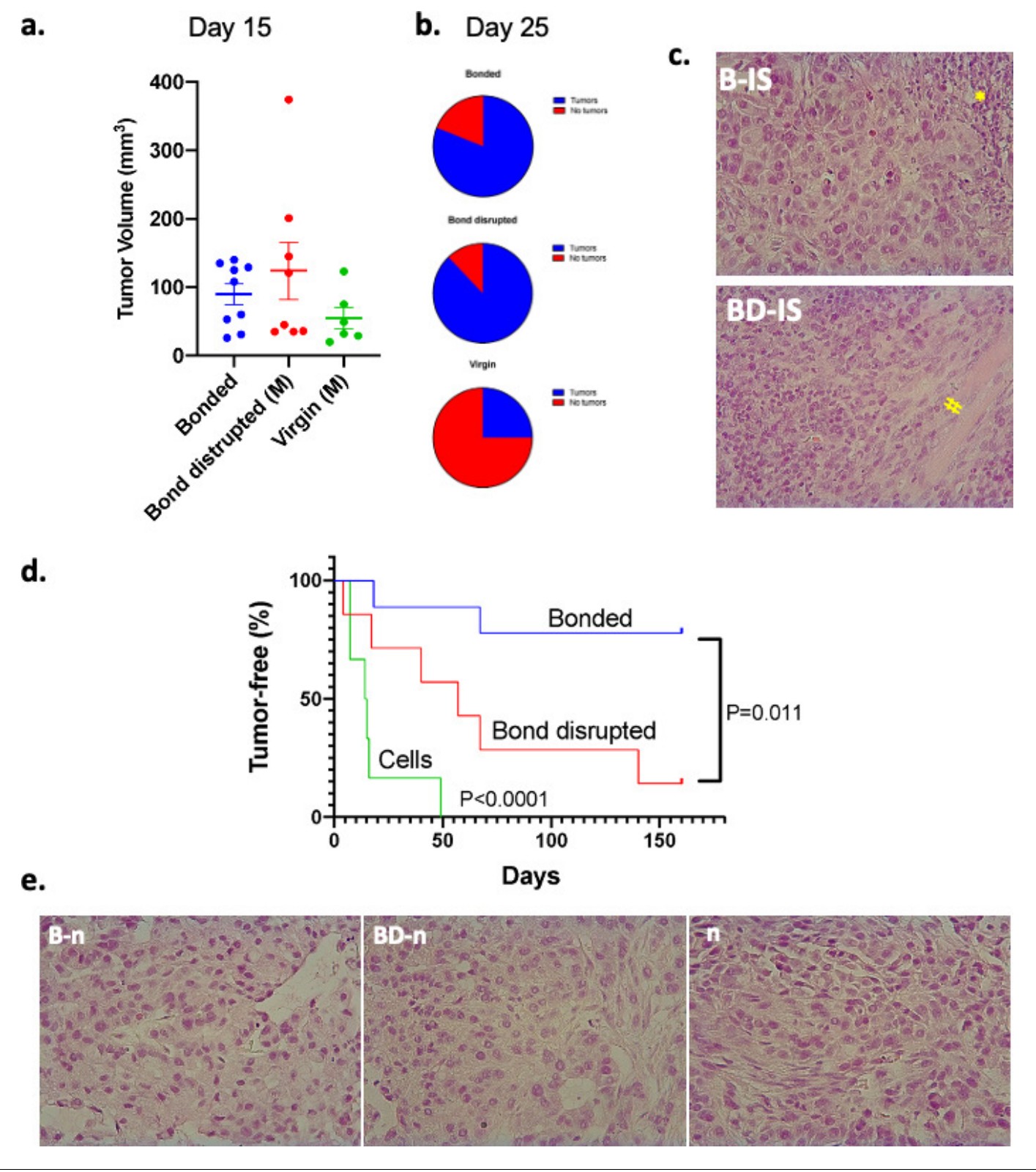

**Figure 2.** Growth of A549 human lung cancers in *P. californicus* (IS stock) and bonding experience. Vasectomized males were used in all studies. (**a**) Volume of measurable tumors at day 15 following cancer cell inoculation in the bonded (n = 9), bond-disrupted (n = 8), and virgin (n = 6) groups; out of the 11, 9 and 8 animals implanted originally with A549 cells. (**b**) Pie graphs indicating the percentage of animals bearing tumors at day 25. (**c**) Representative hematoxylin and eosin (H and E)-stained sections of *P. californicus*-grown tumors from bonded (upper panel), and bond-disrupted

*Figure 2 continued on next page*

*Figure 2 continued*

(lower panel) groups. (*) and (#) indicate necrotic areas and muscle invasion, respectively. (**d**) Tumor-free nude mice implanted with A549 tumor explants from bonded (n = 9) and bond-disrupted (n = 7) *P. californicus*. p-values (log-rank [Mantel–Cox] test) are shown. (**e**) H and E-stained sections of A549 tumors in nude mice derived from explants of A549 tumors from bonded, and bond-disrupted *P. californicus*. The morphology of A549 tumors from the direct inoculation of A549 cells in nude mice is shown (n). N is indicated in the text. B-IS, tumors growing in bonded *P. californicus*; BD-IS, tumors growing in bond-disrupted *P. californicus*; B-n, tumors that originally developed in bonded *P. californicus* and now growing in nude mice; BD-n, tumors that originally developed in bond-disrupted *P. californicus* and now growing in nude mice; n, tumors that developed in nude mice following injection of A549 cells.

SOCS3. Hes1 is a transcription factor that is downstream of Notch signaling, for which the pro-oncogenic activity in lung cancer has been established (*Westhoff et al., 2009*; *Yuan et al., 2015*). NR4A1 encodes for the orphan nuclear receptor A1 for which a strong association with unfavorable outcome in lung cancer has been shown and is involved in cancer cell migration (*Zhu et al., 2017*; *Hedrick et al., 2018*). SOCS3 is a suppressor of cytokine signaling and is a repressor of lung tumorigenesis (*He et al., 2003*; *Lund and Rigby, 2006*). FGG encodes for fibrinogen gamma chain that has been linked to enhanced invasion of lung and other cancer cells (*Sahni et al., 2008*; *Zhang et al., 2019*). The genes that were uniquely detected in the BD vs B groups comparison were FGA and FGB, which encode for fibrinogen A and B chains (*Pieters and Wolberg, 2019*), while in the V vs B comparison, the oncogene Jun that enhances lung cancer cell migration (*Shimizu et al., 2008*) and the connective tissue growth factor that at least in lung cancer, is associated with favorable prognosis (*Chien et al., 2006*; *Chang et al., 2004*). Pathway enrichment analysis indicated that processes associated with differentially expressed genes were linked to the regulation of cell migration and spread, or tissue morphogenesis (*Table 2*).

## Monogamous and not polygamous Peromyscus are sensitive to the effects of bond disruption in spheroid formation

The findings on *P. californicus* prompted us to explore whether other Peromyscus species are also sensitive to the effects of the disruption of pair bonds. Thus, we compared the effects of sera from bonded or bond-disrupted polygamous *P. maniculatus* and monogamous *Peromyscus* polionotus, in the size and shape of A549 tumor spheroids. As shown in *Figure 4*, the disruption of pair bonds altered spheroid morphology in the monogamous, but not in the polygamous species. The intensity of this effect was variable among the animals tested and was recorded in at least 6 of 12 male *P. polionotus* but none of *P. maniculatus* (n = 12) tested (p=0.005, chi-square test; *Figure 4—figure supplement 1*). Contrary to *P. californicus* though, at which pair-bond disruption enhanced spheroid size, in *P. polionotus* the primary effect was seen in the spheroids' shape: Spheroids that formed in the presence of *P. polionotus* sera obtained after the disruption of pair bonds had scattered morphology, as opposed to the spheroids from *P. maniculatus* sera at bonding and bond disruption and those of *P. polionotus* at bonding that were smooth-edged. In some instances (about 25% of animals), this scattered phenotype was also noted in *P. polionotus* sera obtained at bonding (*Figure 4—figure supplement 1*). Whether this difference represents the actual phenotypic difference between the two species or is due to methodological changes in the state of the cells and donor animals remains to be established. In addition, it may reflect the same effect (cell dispersion followed by proliferation) but recorded at different stages during the formation of the spheroids. It is also noted, that the monogamous behavior in Peromyscus has developed independently during the evolution of *P. polionotus* and *P. californicus*, and thus alternative signaling ques may have been engaged in altering the consequences of bond disruption in spheroid formation (*Jašarević et al., 2013*). To that end, the signaling cascades influencing spheroid size and shape may be distinct for the two species; nevertheless, the effects of pair-bond disruption persist.

## Discussion

The present findings exemplify the role of the context – in its wider sense – in cancer progression and underscore the significance of psychosomatic factors as modulators of cancer growth. Using a behaviorally relevant animal model, our results highlight the biological basis of the 'widowhood effects' and suggest that it operates as a tumor-promoting factor, beyond lifestyle changes. Our

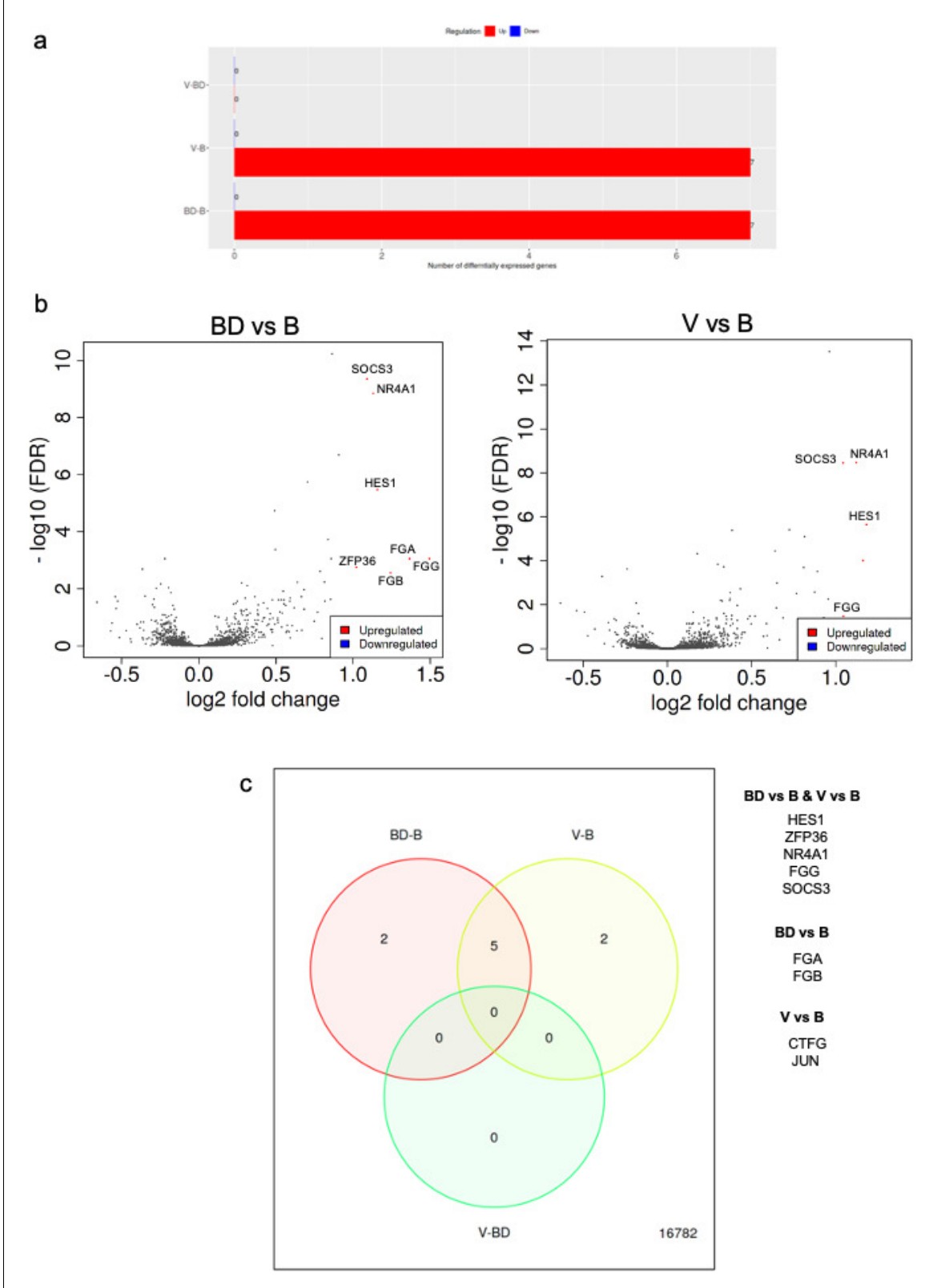

**Figure 3.** RNAseq analysis of A549 cells cultured in the presence of sera from bonded (B), bond-disrupted (BD), or virgin (V) *P. californicus.* (a) Bar graphs showing number of differentially expressed genes in each pairwise comparison group. (b ) Volcano plots showing differentially expressed genes between the B vs BD, and B vs V groups. (c) Venn Diagrams showing overlapping differentially expressed genes. The identity of genes is shown in the
*Figure 3 continued on next page*

*Figure 3 continued*

right. In the bonded group, mice were paired for 12 months. For the bond-disrupted group, we separated paired mice after 12 months of bonding, and collected the sera 1 week after bond disruption. For virgin mice, we collected sera from mice housed 3/cage.

The online version of this article includes the following figure supplement(s) for figure 3:

**Figure supplement 1.** CSC markers in A549 cells cultured in vitro and tumors.
**Figure supplement 2.** Unsupervised hierarchical clustering based on log2 transformed values on all RNAseq data.
**Figure supplement 3.** RNAseq analysis summary of A549 cells.

conclusions are based on the recorded effects of pair bonding in three major phenotypic characteristics of the cancer cells. Those included tumor spheroid formation established in the presence of sera from bond-disrupted animals, the expression profile of the cancer cells in vitro and in vivo that depended on the bonding history of serum donors and tumor hosts, respectively, and ultimately their tumorigenicity in the nude mice. The use of sera from outbred, genetically diverse rodents, allowed us to obtain evidence that this effect varies among individuals but persists across different lung cancer cells. This observation might be of relevance to the study of human populations that are genetically diverse and their responses to the same stimuli may be variable.

In our animal model, cancer cells were implanted in tumor-free animals and the kinetics of tumorigenesis was affected by the animals' bonding history. Whether pair bonding and disruption can also influence tumor initiation will have to be established, nevertheless, the fact that most cancers are slow growing in patients is consistent with the effects of widowhood in influencing the progression, as opposed to the initiation of the disease. Yet, by using the in vivo experiments immunocompromised animals (nude mice and cyclosporine administration), our study suffers from the absence of integration of immune responses that may be especially relevant to widowhood-associated stress.

An unexpected finding was the loss of the tumors in the virgin animals as opposed to the majority of the bonded and bond-disrupted that retained them (*Figure 2b*). A possible explanation is probably related to the differential effectiveness of immunosuppression by cyclosporine. Especially during the initial period after cancer cell inoculation, cyclosporine may have caused more potently immunosuppression in the animals that had been subjected to bonding, due to the concomitant anti-

**Table 1.** Differentially expressed genes between the B, BD, and V groups.
Chromosomal location, fold change (log2), and adjusted p-value are indicated. Genes that are common in the B vs BD and V vs B comparisons are underlined.

| Symbol | Chr | log2 fold change | Adj.pval |
|---|---|---|---|
| BD vs B | | | |
| FGG | 4q32.1 | 1.49 | 8.82e-04 |
| FGA | 4q31.3 | 1.37 | 8.82e-04 |
| FGB | 4q31.3 | 1.24 | 2.78e-03 |
| HES1 | 3q29 | 1.16 | 3.36e-06 |
| NR4A1 | 12q13.13 | 1.13 | 1.42e-09 |
| SOCS3 | 17q25.3 | 1.09 | 4.39e-10 |
| ZFP36 | 19q13.2 | 1.02 | 1.77e-03 |
| V vs B | | | |
| CTGF | 6q23.2 | 1.26 | 5.68e-02 |
| HES1 | 3q29 | 1.18 | 2.26e-06 |
| ZFP36 | 19q13.2 | 1.16 | 9.80e-05 |
| NR4A1 | 12q13.13 | 1.12 | 3.41e-09 |
| FGG | 4q32.1 | 1.08 | 5.39e-02 |
| JUN | 1p32.1 | 1.04 | 3.36e-02 |
| SOCS3 | 17q25.3 | 1.04 | 3.54e-09 |

**Table 2.** Biological processes associated with the differentially expressed genes in B vs BD and V vs B groups (*He et al., 2003*).
The adjusted p-values are indicated.

| Group comparison | Adj.pval | Biological process |
|---|---|---|
| BD vs B | 3.1e-06 | Positive regulation of substrate adhesion-dependent cell spreading |
| V vs B | 4.9e-04 | Blood vessel development |
| | 5.3e-04 | Positive regulation of intracellular signal transduction |
| | 5.3e-04 | Anatomical structure morphogenesis |
| | 5.3e-04 | Regulation of cellular protein metabolic process |
| | 5.3e-04 | Negative regulation of apoptotic process |
| | 5.3e-04 | Positive regulation of cell differentiation |
| | 5.3e-04 | Regulation of epithelial cell proliferation |

inflammatory action of oxytocin, a neurohormone with essential role in the establishment of social interactions and pair bonding (*Lutgendorf et al., 2005*; *Fagundes et al., 2011*; *Fuligni et al., 2009*; *Yuan et al., 2016*; *Carter and Perkeybile, 2018*). It is noted though that the high difference in the tumorigenicity between virgins and the bonded or bond-disrupted animals, renders differential immune suppression unlikely as the sole contributor for this discrepancy.

Differential analysis of gene expression showed that sera from animals at bonding enriched for genes regulating cell migration and spreading, and tissue morphogenesis, features that are consistent with the recorded changes in spheroid morphology. Although for several of the differentially expressed genes, their downregulation, which was seen in the bonded group, was associated with a favorable prognosis, in some cases, it was not. For example, SOCS3 was downregulated in the bonding group, yet it is a tumor suppressor for lung and other cancers (*He et al., 2003*; *Lund and Rigby, 2006*), which may reflect responses related to oxytocin signaling during bonding (*Matarazzo et al., 2012*).

Beyond its effects in the expression of individual genes, the impact of bonding history in transcription was more clearly reflected in the similarity recorded in the transcriptomic profiles of cells cultured in sera from animals with similar bonding experiences. This was especially pertinent to the bonded and bond-disrupted groups. An intriguing possibility is that this is indicative for the lowest rigidity in the transcriptomic profile induced by the serum of virgin animals, as opposed to the changes triggered by the sera of bonded and of bond-disrupted animals that remained more robust.

Collectively, the results provide a mechanistic foundation for the widowhood effect and suggest that the individuals' social, and especially bonding experiences, modify the transcriptome of lung tumors modulating oncogenic activity. As such, they advocate that cancers at widowhood represent a distinct pathological entity that may deserve targeted therapeutic strategies, which should take into consideration social interactions. Thus, preventive measures could be developed to mitigate such pro-oncogenic effects in individuals at bereavement. Whether these findings do occur and at which extent in other monogamous species, including humans, and whether they are applicable to other cancers as well as other pathologies beyond malignancy, remains to be explored. Finally, the present results also raise some concerns regarding the use of conventional animal models and their ability to accurately capture the whole spectrum of the tumorigenic process and the associated host-derived factors.

## Materials and methods

### Animal studies

Genetically diverse male *P. californicus* (stock IS), *P. polionotus* (PO stock), and *P. maniculatus* (BW stock) were obtained from the Peromyscus Genetic Stock Center (Columbia, SC) (RRID:SCR_002769). Mice were all 14–17 months old and were divided into three experimental groups: bonded, bond-disrupted, and virgin. For the tumor inoculation studies, in the bonded group, mice were

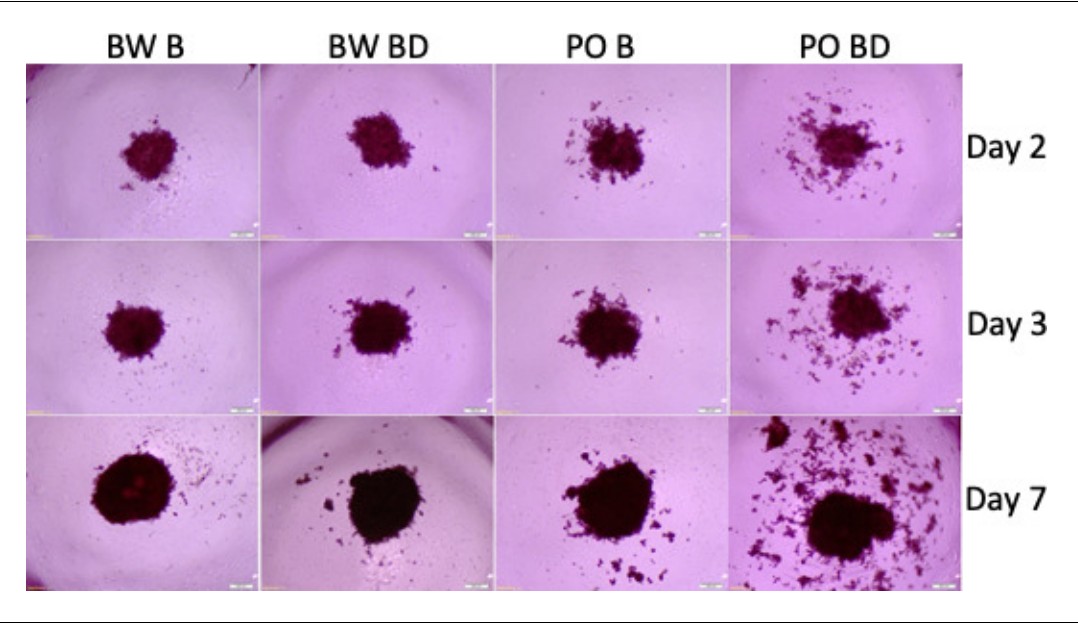

**Figure 4.** Tumor spheroids with sera from *P. polionotus* and *P. maniculatus*. Morphology of tumor spheroids formed after 2, 3, and 7 days in culture with sera isolated from monogamous *P. polionotus* and polygamous *P. maniculatus*. In 6 of 12 *P. polionotus* but none of 12 *P. maniculatus* enhanced dispersion was noted at bond disruption. For the experiment, sera were obtained from the same animal at bonding for 12 months and after 1 week following bond disruption. Bond-disrupted animals were housed independently after separation from females. BW B, *P. maniculatus* bonded; BW BD, *P. maniculatus* bond-disrupted; PO B, *P. polionotus* bonded; PO BD, *P. polionotus* bond disrupted. Epidemiological studies as well as various anecdotal and experiential evidence indicate that social interactions and especially the bonding relationships between couples, influence the development of various pathologies including cancer. A well described consequence of these interactions and of their disruption is the widowhood effect that describes the increased incidence for various diseases in the bereaved partner after loss of a spouse. Such findings, though, are occasionally treated with some skepticism because they are associated with lifestyle changes at widowhood, or homogamy (the likelihood of individuals to marry people of similar health). In the present study, we explored if tumorigenesis is associated with the bonding experience in monogamous rodents (Peromyscus) at which disruption of pair bonds is linked to anxiety and stress. In our studies we compared the formation of lung cancer cell spheroids that formed in the presence of blood sera obtained from bonded and bond-disrupted deer mice. Our studies showed that the disruption of pair bonds altered the size and morphology of spheroids in a manner that is consistent with the acquisition of increased oncogenic potential. In vivo experiments were also performed consisting of the consecutive transplantation of human lung cancer cells between monogamous Peromyscus californicus, differing in bonding experiences and nude mice. These studies showed that bonding suppressed tumorigenicity in nude mice suggesting that the protective effects of pair bonds persisted even after bonding ceased. Differential gene expression analysis at the whole transcriptome level, indicated changes in cell adhesion and migration between cells cultured with sera from bonded and bond-disrupted or virgin animals. Furthermore, the RNA expression profile of cancer cells that had been cultured in the presence of Peromyscus serum predicted the bonding history of the serum donors. The results highlight the role of the context - in its wider sense - in cancer progression and underscore the significance of psychosomatic factors as modulators of cancer growth. Our analyses provide a mechanistic foundation for the widowhood effect and suggest that the individuals' social, and especially bonding experiences, modify the transcriptome of lung tumors modulating oncogenic activity.

The online version of this article includes the following figure supplement(s) for figure 4:

**Figure supplement 1.** Microsphere morphology of A549 human lung cancer cells cultured in *P. maniculatus* or *P. polionotus sera*.

---

paired for at least 2 months before the study began and remained paired until the end of the study. In the bond-disrupted group, mice were paired 2 months before the study started, and immediately after cancer cells injection, they were separated. In the virgin group, mice were kept individually 2 months before the study began. Vasectomy was performed to prevent pregnancy during the study. Some siblings were used and were distributed randomly in different experimental groups as

described in the legend of *Figure 1—figure supplement 1*. Nude mice (male, 6–8 weeks old) were obtained from Charles River Laboratories (Boston, MA) and were housed in groups of 4–5. For serum collection used in the RNAseq studies and spheroid formation, for the bonded group, mice were paired for about 12 months. For the bond-disrupted group, we separated paired mice after 12 months of bonding and collected the sera 1 week after bond disruption. For virgin mice, we collected sera from mice housed 3/cage. Sera were obtained by retro-orbital bleeding before and after bond disruption at the indicated times. Animal studies were approved by the University of South Carolina IACUC (Protocol # 2473-101464-102319).

## Cell lines

A549 human non-small cell lung adenocarcinoma cells were originally obtained from ATCC (Manassas, VA) and thereafter maintained in freezing media (60% Dulbecco's modified Eagle medium [DMEM], 30% FBS, 10% dimethyl sulfoxide). Most recently, cells were validated by STR typing (Biosynthesis, Lewisville, TX) just after completion of experiments. Human H1703 squamous, H596 adenosquamous, H358 bronchioalveolar, and H292 mucoepidermoid lung cancer cells were obtained prior to their use from ATCC (Manassas, VA) and cultured for three passages at ATCC-recommended media prior to the performance of the spheroid assays. Cells were tested negative for mycoplasma contamination.

## Tumor inoculation

To cause immunosuppression in *P. californicus* and overcome xenograft rejection, animals were treated daily with 100 mg/kg cyclosporine A (in 90% olive oil and 10% EtOH) s.c. starting 1 day before the implantation of cancer cells, for 2 weeks, and then every other day for the whole duration of the study (*Perea-Rodriguez et al., 2015*). For cancer cell inoculation, ($5 \times 10^6$) cells were injected subcutaneously into the right flank of mice in a total volume of 100 µl phosphate-buffered solution (PBS). Tumor volumes were assessed by using the following formula: (width)$^2 \times$ length/2. All experiments were approved by the Institutional Animal Care and Use Committee of the University of South Carolina (approval no. 101464). For re-transplantation in nude mice, tumors were harvested from *P. californicus*, mechanically minced at pieces of 5–10 mm$^3$, and were implanted into the right flank of nude mice using a trocar needle. Mice were followed up each week until 4 months.

## Histology

Tumor was fixed in 4% neutral buffered formalin and subsequently embedded in paraffin. Sections were stained with hematoxylin and eosin for histological assessment. Where available, a part of the tumor was snap-frozen on dry ice and stored at −80℃, for RNA extraction. Images were obtained by a Leica optical microscope.

## Tumor spheroid formation

Lung cancer cells were seeded into 96-well spheroid microplates (Corning Cat. No. 4515) at $2 \times 10^3$ cells/well in 100 µl of DMEM+5% FBS+5% serum of each mouse. The age of the mice, their bonding group, and the period of bonding are described in the text and corresponding figure legends. The plate was incubated at 37℃, 5% $CO_2$. Images were taken using an inverted microscope at 4× magnification each day until 3 days and analyzed using NIH ImageJ software to assess microsphere areas and volumes. The studies were repeated independently at least twice, and similar results were obtained. For the assessment of the spheroids that formed with *P. polionotus* sera, 'scattered' phenotype was scored when at least two outgrowths formed distal from the main spheroid.

## Cell viability assay

Spheroid cell viability was assayed using the LIVE/DEAD Viability/Cytotoxicity Kit (Cat. No. L3224). After 3 days of spheroid culture, wells were rinsed two times with an 80 percent-volume change of media with D-PBS. EthD-1 (12 µM) and calcein AM (4 µM) were added to the wells, and the cells were incubated in the dark for 30–45 min to avoid the photodynamic effect. Images were taken using a fluorescence microscope; live cells fluoresce green, whereas dead cells fluoresce red. Data were analyzed using ImageJ image analysis software.

## Quantitative real-time PCR analysis and RNA sequencing

Total RNA from cell and tumor tissues were isolated using the Qiagen RNeasy Mini Kit. Equal quantities of RNA were used for making cDNA using iScript cDNA synthesis kits (Bio-Rad) according to the supplier's protocol on a T100 thermal cycler (Bio-Rad). Human-specific primers for CSC-related genes: OCT4, β-catenin, and CD133 were designed using Primer3 and Primer BLAST. Quantitative real-time PCR was performed using the Bio-Rad Real-Time PCR detection system and iTaq Universal SYBR Green Supermix (Bio-Rad) according to the manufacturer's instructions. Amounts of target genes mRNA were normalized to a reference gene GAPDH and were expressed as arbitrary units. The oligonucleotides used for qPCR amplification were as follows: Oct-4: GAAGGATGTGGTCCGAGTGT (left) and GTGAAGTGAGGGCTCCCATA (right); b-catenin: GAAACGGCTTTCAGTTGAGC (left) and CTGGCCATATCCACCAGAGT (right); CD-133: TTGTGGCAAATCACCAGGTA (left) and TCAGATCTGTGAACGCCTTG (right); GAPDH: CCATCACCATCTTCCAGGAGCG (left) and AGAGATGATGACCCTTTTGGC (right). Hierarchical clustering analysis and presentation of expression data were performed using the Morpheus analysis software (https://software.broadinstitute.org/morpheus). For the analysis, raw cpm values were either used or transformed by using the formula Log2 (1 + raw values), as described in the text. RNA sequencing was performed as described (*Chavez et al., 2020*). RNAseq data were deposited to NCBI (GSE167827). Differential analysis of gene expression and enrichment pathway analysis were performed by using the iDEP platform (*He et al., 2003*).

## Statistical analysis

The data are presented as mean ± SEM. Statistical analysis was performed by paired t-test, chi-square test, ANOVA, or log-rank (Mantel–Cox) test as indicated in the figure legends and text. Results were considered significant when $p \leq 0.05$. All graphs were generated using GraphPad Prism software (version 8).

## Acknowledgements

The results shown here are in part based upon data generated by the TCGA Research Network: https://www.cancer.gov/tcga. We thank Hao Ji and Dr Michael Shtutman form the UofSC Functional Genomics Core for the RNA sequencing analysis and Dr Vitali Sikirzhytski for help with fluorescent imaging. This study was supported by NSF (Award Number: OIA1736150).

## Additional information

### Funding

| Funder | Grant reference number | Author |
|---|---|---|
| National Science Foundation | OIA-1736150 | Hippokratis Kiaris |

The funders had no role in study design, data collection and interpretation, or the decision to submit the work for publication.

### Author contributions

Asieh Naderi, Data curation, Formal analysis, Investigation, Methodology, Writing - original draft, Writing - review and editing; Elham Soltanmaohammadi, Data curation, Investigation, Writing - review and editing; Vimala Kaza, Resources, Investigation, Writing - review and editing; Shayne Barlow, Investigation, Writing - review and editing; Ioulia Chatzistamou, Conceptualization, Investigation, Writing - review and editing; Hippokratis Kiaris, Conceptualization, Formal analysis, Supervision, Funding acquisition, Writing - original draft, Project administration, Writing - review and editing

### Author ORCIDs

Hippokratis Kiaris (iD) https://orcid.org/0000-0002-8999-8289

## Ethics

Animal experimentation: University of South Carolina IACUC (Protocol # 2473-101464-102319).

# Additional files

## Supplementary files
• Transparent reporting form

## Data availability

Peromyscus animals are available from the Peromyscus Genetic Stock Center.

The following dataset was generated:

| Author(s) | Year | Dataset title | Dataset URL | Database and Identifier |
|---|---|---|---|---|
| Naderi A, Ji H, Michael S, Kiaris H | 2021 | Whole Transcriptome RNA-seq from A549 cells exposed to P. californicus serum | https://www.ncbi.nlm.nih.gov/geo/query/acc.cgi?acc=GSE167827 | NCBI Gene Expression Omnibus, GSE167827 |

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
