## [Decision Letter]

**Acceptance summary:**

This study provides evidence that pair bonding may have a favourable role in cancer. To this end, the authors show that disruption of pair bonds in monogamous rodents results in faster growth of ectopic tumors formed by various lung cancer cell lines. Moreover, evidence is provided that sera from paired and pair-disrupted animals differentially affect gene expression in cancer cells, whereby sera from pair-disrupted animals stimulate growth of cancer cells more strongly than the sera from paired animals. Overall, these findings underscore potentially significant impact of psychosomatic factors caused by changes in social environment on tumor growth rates. More broadly, these results emphasize importance of considering social environment in clinical management of neoplasia.

**Decision letter after peer review:**

Thank you for submitting your article "Persistent effects of pair bonding in lung tumorigenesis in monogamous rodents" for consideration by *eLife*. Your article has been reviewed by 3 peer reviewers, and the evaluation has been overseen by a Reviewing Editor and a Senior Editor. The following individual involved in review of your submission has agreed to reveal their identity: William W Lockwood (Reviewer #2).

Summary

In this study, Naderi et al provide evidence for beneficiary effects of pair bonding in neoplasia by showing that disruption of pair bonds in monogamous rodents accelerates growth of ectopic tumors formed by A549 lung cancer cells. Moreover, the authors provide evidence that sera from pair-disrupted animals bolster A549 spheroid growth. Overall, it was found that the observations reported in this study are of high potential interest inasmuch as they emphasize potential importance of social environment and associated psychosomatic factors (e.g. anxiety, stress) in modulating tumor growth. However, lack of mechanistic evidence linking pair bonding to tumor growth decreased the enthusiasm for the study. Some methodological limitations and issues with insufficient description of experimental procedures were also noted.

Essential Revisions:

1. It was thought that the major limitation of the study is that it relies on a single cell line (A549) which brings to question the generality of the conclusions. To this end, it was found that this study would significantly benefit from using additional cancer cell lines and/or carcinogens (to study effects of pair bonding on tumor initiation). Alternatively, the authors should clearly indicate this limitation and accordingly modify their conclusions as well as the title and abstract of the manuscript.

2. Another major issue that was observed is the apparent lack of mechanisms explaining the effects of bonding on tumor growth. Solely relying on cancer stem cell markers was found to be insufficient to render clustering analyses informative and support authors conclusion. Pertinent to this, additional factors and/or parameters of cancer growth should be included. It was suggested that performing gene expression profiling in this context may be warranted as such approach is likely to illuminate the molecular underpinnings of observed phenomena.

3. It was found that in some experiments (e.g. transplantation studies), the sample sizes were on the modest side. Sample size and statistical power information should be provided throughout the manuscript (including the abstract). Finally, it was deemed that the methodology was not sufficiently explained and that experimental details were missing. For instance, the authors should provide information of the number of replicates.

This summary was based on individual Reviewers' assessments (provided below) and subsequent discussion.

*Reviewer #1:*

This is an inventive paper at the intersection of the social and the biological sciences – using experimental methods in an animal model almost perfectly suited to test the ideas in question. The topic is whether the disruption of pair bonding actually can facilitate tumorigenesis, and the investigators use several in vitro and in vivo approaches, and several rodent species, to show that this is the case. The results comport with epidemiological studies that have shown that recently widowed individuals have higher risk of various diseases, including cancer.

I found this paper to be inventive, at the intersection of the social and the biological sciences, using experimental methods in an animal model, almost perfectly suited to test the ideas in question. The topic is whether the disruption of pair bonding actually can facilitate tumorigenesis, and the investigators use several approaches, comporting with epidemiological studies that have shown that recently widowed individuals have higher risk of various diseases, including cancer.

One of the things I would have liked to of seen is more attention to the temporal dynamics, including how soon after disruption of the bond there was evidence of increased tumor risk. In humans, most cancers are slow growing, and one of the issues with studies of the widowhood effect is that while it makes sense that the loss of a spouse might lead to the sudden, new onset of a stroke, say, it's hard to imagine how it can prompt the rapid emergence of colon cancer. My point is that it seems more likely that, say, a small and nascent tumor has its course accelerated, rather than initiated, by the loss of a partner. Hence, an animal model of this might involve first implanting a tumor into pair-bonded rodents, and then disrupting the pair bond for some of them and assessing how the course of illness was modified. I realize this is a different set of experiments, and I am not suggesting that these must be done at present, since the current results are themselves a very meaningful advance, in my view.

I think the abstract should include samples sizes. In addition, in the abstract, the authors might want to mention that another possible explanation for why the risk of death from cancer is higher in people who are recently widowed is the potential for homogamy based on cancer risk. In other words, imagine that people in poor health are more likely to marry people in poor health (which is true), then people with similar cancer risk would marry each other and the death of the first partner would simply be an indicator of these unknown factors predisposing to the union and then the subsequent partner would also die of cancer.

In the Results section, such as a page 4, I would include the sample size of all analyses even though the details are reported in the methods section.

Similarly, although this is mentioned in the methods section, I would make it clear at page 5 that all the mice are of the same age regardless of their mating status. We obviously want to avoid a conflation of mating status with age.

The result in the middle of page 5, which showed that, by day 25, the tumors persisted in both a bond-disrupted and bonded animals, at 89 and 82% respectively, while in virgin animals there were detectable in only 25% of cases, was quite perplexing. The authors speculate that this may have to do with concomitant anti-inflammatory action of oxytocin, but the size of the difference is so large that I was left wondering what could possibly be going on.

At page 7, I wondered how big the sample sizes were in ascertaining that certain tumors from the bonded animal groups retained a distinct profile. And, analytically, it seems the investigators clustered all three of their kinds of experiments into one hierarchical model. I'm not sure that is optimal.

Also at page 7, the paper suddenly mentioned that some of their samples were siblings. I suspect the statistical controls are adequate but I was left wondering why, methodologically, they didn't simply exclude siblings from the animals they used in their experiments?

I thought the results from the monogamous and polygamous mice were especially strong and revealing.

Overall, I thought this was a very inventive paper with many novel results that open up new frontiers in thinking about not only the underlying biosocial phenomena, but also how to experimentally manipulate and analyze them.

*Reviewer #2:*

Naderi et al. present an interesting short report that aims to experimentally assess the influence of pair bonding on tumorigenesis. While epidemiological studies have suggested that social interactions and the bond between couples affects tumor risk and development, this is been difficult to independently assess in humans due to potentially confounding issues associated with other lifestyle variables. Thus, the underlying biological processes regulating the observed protective effects of marriage and the adverse events correlated with the of the loss of a partner are still largely unknown. Using monogamous rodents as a model system, the authors aim to address this question and directly test the impact of pair bonding and disruption on the tumorigenic potential of human lung cancer cells. They find that sera from pair disrupted Peromyscus californicus increases the size and morphology of A549 cells grown as spheroids, while sera bonded mice had no effect compared to sera from non-pair bonded animals. Importantly, these effects were also seen with sera from P. polionotus, which is also monogamous, but not from the polygamous P. manicultus where no effects of pair disruption were observed. This effect was also demonstrated in vivo when A549 cells were transplanted into pair bonded, disrupted or virgin animals or from these animals into nude mice, suggesting the long-lasting effects on tumorigencity of the cells.

Overall, the study is innovative for its use of monogamous rodent species for the study of the long outstanding question of the biological effects of social interactions on cancer development. As the most common model organism for the study of cancer, *Mus musculus*, is promiscuous and does not form long-term pair bonds, these results suggest that important factors regulating cancer development may be lacking in many preclinical studies. This is an important consideration and may influence how studies are modeled in the future to capture additional host aspects associated with tumorigenesis. The authors ultimately succeed in demonstrating that pair bonding should be considered in this regard as their in vitro experiments, while limited in breadth, do indicate clear effect of sera from pair disrupted animals on tumorigenic potential across monogamous, and not polygamous, species. While the authors do not comprehensively assess the underlying processes regulating these phenotypes, they demonstrate that cells/tumors from pair disrupted animals cluster together based on the expression of a limited set of cancer stem cell markers. Together, the authors present sufficient in vitro and in vivo data to conclude that pair-bonding affects tumor cell behaviour.

While unique in design and informative in its overall conclusions, it should be noted that the study does have many important limitations. First, only a single lung cancer cell line (A549) is used in the study and it is difficult to determine whether the observed findings will be generalizable across different tumor types (aside from lung adenocarcinoma) or those with different mutation spectra and driver oncogenes (aside from KRAS G12S). Furthermore, limited mice are used in transplantation experiments and it is also unclear whether there is substantial diversity in the effects from sera from different animals. The number of animals used to isolate sera is not indicated and therefore, it is unclear how variable the effects on spheroid growth are and whether this influences the conclusions.

Another limitation is the use of cancer cell lines. These cells are already tumorigenic and pair disruption in this instance is therefore testing whether there is an influence on making these cells more or less tumorigenic. This is different than the real-life scenario where it is likely that spouses of deceased partners will not have cancer in the first place, but instead, may develop cancer as a result of the stress and other factors associated with pair disruption.

The assessment of only select cancer stem cell markers is an additional limitation. While these are no-doubt important for influencing tumorigenic potential, they represent only a fraction of the changes that could be occurring as a result of pair bonding disruption. As such, this analysis is very limited in breadth. Furthermore, it is difficult to assess the clustering results as the limited variables tested impacts the ability to discriminate between samples. Thus, this type of analysis isn't the most appropriate as currently presented and will likely require the assessment of additional variables to accurately separate the samples by underlying phenotypes.

Another limitation is the use of cancer cell lines. These cells are already tumorigenic and the influence of pair disruption in this instance is therefore testing whether it can make these cells more or less tumorigenic. This is different than the real-life scenario where it is likely that spouses of deceased partners will not have cancer in the first place, but instead, may develop cancer as a result of the stress and other factors associated with pair disruption. Thus, a more appropriate and complementary model would assess the effects of pair disruption on cancer initiation as opposed to development and progression. For example, pair bonded and disrupted animals could be given a carcinogen to assess the rate and incidence of tumor onset. Or, cell lines that represent a pre-malignant state (ie. don't form tumors in nude mice) could be used to assess the effects on progressing these cells to a fully malignant state.

*Reviewer #3:*

The work provides an interesting observation that the bonding between monogamous rodents may help in diminishing the ectopically inoculated tumour size and a disruption of this pair bonds resulted in increased tumour size probably due to anxiety and stress. This paper will be of interest to support psychosomatic medicine in cancer treatments. The authors used sera collected from three groups (virgin, bonding, and disruption of bonding) monogamous rodents to evaluate the effect on tumour cells growing in spheroids. Tumour cells were also subcutaneously inoculated into flank of the three groups of mice to evaluate the actual tumour growth.

Strengths:

An interesting approach to use both tumour cells grown on spheroids and in mice with different pairing bonding conditions. It was relatively convincing results that tumour cells grow faster in a condition associated with bonding disruption.

Weaknesses:

Although the paper has strengths in principle, the weaknesses of the paper are that these strengths are only reflected in a descriptive manner. Epidemiological evidence already suggests that social interactions and especially bonding between couples influence tumorigenesis. The observation in this paper is interesting and confirmatory of cancer patients but not novel. It would have been informative had the authors utilised the models to perform a more in-depth investigation in providing mechanistic insight on how the bonding disruption affects tumour growth. Some experimental data and experimental details require clarification to allow confident interpretation of the described results. The brief discussion was presented mostly as a summary and statement without sufficient literature support and citation, which should be expanded and enhanced.

The manuscript submitted by Naderi et al. entitled "Persistent effects of pair bonding in lung tumorigenesis in monogamous rodents" includes interesting observations that pair-bonding between monogamous rodents (P. californicus) is beneficial to the mice in preventing the implanted tumor growth. However, the current version of the submitted manuscript includes the only limited scope and lack of mechanistic insight. Multiple experimental details were missing that prevent a confident interpretation of the described results.

1. In figure 1b, it is not clear when the serum from the bonding (B) group was collected (only one B condition was shown). The appropriate control should include serum from the bonding group at all time points (24h, 1week, and 2 weeks) as the bonding disruption (BD) group for a direct comparison.

2. There is a mistake in figure 2b. How could it be possible that the 6 mice with the measurable tumor in the virgin (V) group only accounted for 25% of the tumor-bearing sub-group in the pie chart when the total mice number was 8? Based on the results in Figure 2a, the mouse number with a tumor should be 6 and accounted for 75% (6/8) of the total virgin group. Page 5, sentence 99-105 should be better included in the discussion than in the Results section. In addition, the difference is only between 82% (B) and 75% (V), rather than the described 25%, which did not seem to be significant.

3. In figure 2d, how many tumors were transplanted? How many nude mice were used? The information was not included in the methods, nor in the results or figure legends. All of the "N" should be clearly described in the figure legends. Were all tumor cells isolated from all 9 tumor-bearing mice in the B group and 8 tumor-bearing mice from the bonding disruption (BD) group dissected and used for transplantation study? How many different tumor cells from each group (V, B, and BD) were transplanted into nude mice? How many nude mice were transplanted with tumor cells from each tumor-bearing mouse?

4. It is not clear how many times the in vivo animal experiments were performed? Did the in vivo data showing in figure 2 represent only a one-time experiment?

5. In figure 3, the signature using only three genes was overly simplified. The selected markers did not provide supporting information on why serum from the different mice groups displayed various sizes in spheroids and tumors because the gene expression of the BD groups was consistently located in between B and V groups, which all showed smaller tumor sizes.

6. The clustering description of figure 3 was confusing and the signature was not unique, which was not unexpected because the expression data of these genes did not significantly differ among the different groups. Non-biased analysis using sequencing results with unsupervised clustering likely will provide data that are more informative and mechanistic.

7. The sentence from lines 127-130 is an over-interpretation of the results. The description of transcription could be better included in the discussion and not in the Results section.

8. In figure 4, the description of the serum collection was absent. When was the sera collection time? Were the polygamous mice housed individually after disruption (probably yes) and separated from all-female mice. The figure legend was incomplete and lack important details. The abbreviation should also be described.

9. The data in figure 4 regarding the effects on different shapes of spheroids (but without clear biological consequence) did not positively support the hypothesis. Additional mouse species may be needed to identify if the tumor size difference observed in P. californicus after bonding disruption was an exception or a general observation. Alternatively, the authors should limit the claim and re-write the title to reflect the observation in the specific mouse strain.

10. In figure 4, the serum collected from the "same animal" at bonding and following the bond disruption did not take into consideration of aging and hormonal effects. Additional groups of animals should have been included to address these concerns and avoid the over-interpretation of the results.

11. The brief discussion was presented mostly as a summary and statement without sufficient literature support and citation, which should be expanded and enhanced.

---

## [Author Response]

Essential Revisions:1. It was thought that the major limitation of the study is that it relies on a single cell line (A549) which brings to question the generality of the conclusions. To this end, it was found that this study would significantly benefit from using additional cancer cell lines and/or carcinogens (to study effects of pair bonding on tumor initiation). Alternatively, the authors should clearly indicate this limitation and accordingly modify their conclusions as well as the title and abstract of the manuscript.

We agree with this comment of the reviewers and editors. To address this, we used additional lung cancer cell lines (H1703, H596, H358, and H292). In the absence of animals used originally that were 14-17 months old and bonded for 1 year or more, younger animals (6-8 months old) bonded for 2 months were now used. Not only we were able to record for all cells lines similar effects with analogous variation as to that recorded originally (now shown as scatter dot plots in original Figure 1c), but more importantly we think, we were able to show that the effect varies among individuals and is produced in the different lung cancer cells in a similar manner (Figure 1—figure supplement 1).

2. Another major issue that was observed is the apparent lack of mechanisms explaining the effects of bonding on tumor growth. Solely relying on cancer stem cell markers was found to be insufficient to render clustering analyses informative and support authors conclusion. Pertinent to this, additional factors and/or parameters of cancer growth should be included. It was suggested that performing gene expression profiling in this context may be warranted as such approach is likely to illuminate the molecular underpinnings of observed phenomena.

We agree with the reviewers that the qPCR data on selected cancer stem cell markers are insufficient to support the clustering findings we described in the original manuscript. In this revised version of our manuscript, we included whole transcriptome RNAseq results that confirmed that cells cluster together, depending on the bonding history of the serum donors. In addition, differential expression analysis was performed which pointed to bonding-dependent effects in cell adhesion and migration as well as tissue morphogenetic processes. In the revised version of the manuscript, to accommodate these new results, extensive changes were performed in the abstract, the discussion, the Results section and the figures. We also moved the original qPCR data of cancer stem cell markers into the supplementary information which now, also includes the new clustering data from RNAseq (Figure3 and Figure 3—figure supplement 2,3).

3. It was found that in some experiments (e.g. transplantation studies), the sample sizes were on the modest side. Sample size and statistical power information should be provided throughout the manuscript (including the abstract). Finally, it was deemed that the methodology was not sufficiently explained and that experimental details were missing. For instance, the authors should provide information of the number of replicates

In compliance with the reviewers’ comments details about statistical power, number of replicas and animal numbers in tumor transplantation studies are now mentioned throughout the revised manuscript.

This summary was based on individual Reviewers' assessments (provided below) and subsequent discussion.Reviewer #1:[…] One of the things I would have liked to of seen is more attention to the temporal dynamics, including how soon after disruption of the bond there was evidence of increased tumor risk. In humans, most cancers are slow growing, and one of the issues with studies of the widowhood effect is that while it makes sense that the loss of a spouse might lead to the sudden, new onset of a stroke, say, it's hard to imagine how it can prompt the rapid emergence of colon cancer. My point is that it seems more likely that, say, a small and nascent tumor has its course accelerated, rather than initiated, by the loss of a partner. Hence, an animal model of this might involve first implanting a tumor into pair-bonded rodents, and then disrupting the pair bond for some of them and assessing how the course of illness was modified. I realize this is a different set of experiments, and I am not suggesting that these must be done at present, since the current results are themselves a very meaningful advance, in my view.

We completely agree with the reviewer that indeed, bond disruption most likely accelerates the development of a preexisting condition, rather than initiating it. This is also supported by our model at which the cancer cells were implanted in the animals, just prior to the disruption of the pair bonds. This point is now discussed in the Discussion by the addition of the following sentence:

Discussion (2^nd^ paragraph): “…In our animal model, cancer cells were implanted in tumor-free animals and the kinetics of tumorigenesis were affected by the animals’ bonding history. Whether pair bonding and disruption can also influence tumor initiation will have to be established, nevertheless, the fact that most cancers are slow growing in patients is consistent with effects of widowhood in influencing the progression, as opposed to the initiation, of the disease.”

The temporal dynamics point, is also of major significance. Some hints related to this are provided in the spheroid assays at which sera collected at different time points are used. We have initiated such studies to describe in higher detail these dynamics, but the genetic heterogeneity of the deer mice requires high numbers of animals to accurately describe these differences. We hope that we will be able to describe them in a future publication. In the meanwhile, however we included in this paper as new Figure 1—figure supplement 1, the results of a study involving 7 younger (about 6-8 months old as opposed to the original study that involved 14-17 months old animals) that were bonded for 2 months (as opposed to 12 months of the original study) or bond-disrupted. In this study we found similar effects in spheroid formation with those reported originally. More importantly we obtained evidence that the effect varies among individuals but persists across different lung cancer cells.

I think the abstract should include samples sizes. In addition, in the abstract, the authors might want to mention that another possible explanation for why the risk of death from cancer is higher in people who are recently widowed is the potential for homogamy based on cancer risk. In other words, imagine that people in poor health are more likely to marry people in poor health (which is true), then people with similar cancer risk would marry each other and the death of the first partner would simply be an indicator of these unknown factors predisposing to the union and then the subsequent partner would also die of cancer.

We thank the reviewer for his/her insightful comment. Sample sizes for tumor re-transplantation are now shown in the abstract. The possibility of homogamy as an additional factor explaining the widowhood effect is now described in the abstract.

Abstract (line 2): “…lifestyle changes, homogamy (likelihood of individuals to marry people of similar health) or directly associated with…”

In the Results section, such as a page 4, I would include the sample size of all analyses even though the details are reported in the methods section.Similarly, although this is mentioned in the methods section, I would make it clear at page 5 that all the mice are of the same age regardless of their mating status. We obviously want to avoid a conflation of mating status with age.

Sample sizes and the age of the animals are now included in the Results section of the revised manuscript.

The result in the middle of page 5, which showed that, by day 25, the tumors persisted in both a bond-disrupted and bonded animals, at 89 and 82% respectively, while in virgin animals there were detectable in only 25% of cases, was quite perplexing. The authors speculate that this may have to do with concomitant anti-inflammatory action of oxytocin, but the size of the difference is so large that I was left wondering what could possibly be going on.

We agree with the reviewer that this is really puzzling. We moved the corresponding part in the discussion (paragraph 3) and also added the following at the end:

“…It is noted though that the high difference in the tumorigenicity between virgins and the bonded or bond-disrupted animals, renders differential immune suppression unlikely as the sole contributor for this discrepancy.”

At page 7, I wondered how big the sample sizes were in ascertaining that certain tumors from the bonded animal groups retained a distinct profile. And, analytically, it seems the investigators clustered all three of their kinds of experiments into one hierarchical model. I'm not sure that is optimal.

We agree that clustering analysis, especially as described in the original version of our manuscript suffers from various limitations. Therefore, it is not discussed in detail and the results are moved in the supplementary material (Figure 3-Figure Suppl. 1) in the revised manuscript. However, unsupervised clustering was also performed with the RNAseq data which showed similar trends (Figure 3-Figure Suppl. 2). This analysis showed that expression profiles of bonding and disruption are more robust than those of virgins, as discussed in the revised manuscript (Author response image 1).

**Author response image 1. respfig1:** Expression signature of human A549 lung cancer cells cultured in sera from deer mice differing in bonding history.

We refrained however from describing these results explicitly here, as various analyses are still ongoing and their description is deviating from the original scope of the present study. We wanted to note that by using these RNAseq data (n=6 individuals per group) we were able to define a bonding signature consisting of 15 genes that showed that the discriminatory effects of bonding are more potent than those of disruption (bond-disrupted animals cluster together with virgins while bonded deviate earlier). More surprisingly, this signature predicted lung cancer survival of human patients (Author response image 2).

**Author response image 2. respfig2:** Bonding signature and patients’ prognosis. a. Hierarchical clustering of lung cancer patients from TCGA based on the 13 genes signature and survival. b. Survival probability of the two groups of TCGA lung cancer patients that emerged by using the 13 gene signature.

Also at page 7, the paper suddenly mentioned that some of their samples were siblings. I suspect the statistical controls are adequate but I was left wondering why, methodologically, they didn't simply exclude siblings from the animals they used in their experiments?

The animals are genetically diverse (outbred) and a strategy we occasionally use to minimize potential founder effects is to use siblings distributed in different groups. We tried to do this here but unfortunately, tumor take rates and availability of animas did not allow us to do this rigorously. To avoid confusion and unjustified speculations we now eliminated this part of the discussion (along with the related discussion of clustering) and mentioned in the Methods the sporadic use of siblings (pointing to Figure 1-Figure Suppl. 1) at which mating numbers of the crosses are shown.

I thought the results from the monogamous and polygamous mice were especially strong and revealing.Overall, I thought this was a very inventive paper with many novel results that open up new frontiers in thinking about not only the underlying biosocial phenomena, but also how to experimentally manipulate and analyze them.Reviewer #2:[…] Weaknesses:While unique in design and informative in its overall conclusions, it should be noted that the study does have many important limitations. First, only a single lung cancer cell line (A549) is used in the study and it is difficult to determine whether the observed findings will be generalizable across different tumor types (aside from lung adenocarcinoma) or those with different mutation spectra and driver oncogenes (aside from KRAS G12S). Furthermore, limited mice are used in transplantation experiments and it is also unclear whether there is substantial diversity in the effects from sera from different animals. The number of animals used to isolate sera is not indicated and therefore, it is unclear how variable the effects on spheroid growth are and whether this influences the conclusions.

We agree with the reviewer for the various limitations he pointed out. In this revised version of the manuscript these points were addressed. Specifically, additional lung cancer cell lines were used to explore how general our findings are. The diversity point of the experimental animals, indeed represents an advantage but also a disadvantage of this model. Although randomly animals were assigned into groups, founder effects remain formally possible. Sample numbers, besides legends are also now mentioned throughout the manuscript as well.

Another limitation is the use of cancer cell lines. These cells are already tumorigenic and pair disruption in this instance is therefore testing whether there is an influence on making these cells more or less tumorigenic. This is different than the real-life scenario where it is likely that spouses of deceased partners will not have cancer in the first place, but instead, may develop cancer as a result of the stress and other factors associated with pair disruption.

This is a very important point that we tried to address in the revised manuscript by adding a relevant comment in the Discussion (2^nd^ paragraph). See also response to reviewer 1. Briefly, we agree that our experimental set up addresses the effects of bonding and disruption in preexisting cancers that now become more aggressive after the disruption of pair bonds.

The assessment of only select cancer stem cell markers is an additional limitation. While these are no-doubt important for influencing tumorigenic potential, they represent only a fraction of the changes that could be occurring as a result of pair bonding disruption. As such, this analysis is very limited in breadth. Furthermore, it is difficult to assess the clustering results as the limited variables tested impacts the ability to discriminate between samples. Thus, this type of analysis isn't the most appropriate as currently presented and will likely require the assessment of additional variables to accurately separate the samples by underlying phenotypes.

As we already mentioned we agree that clustering analysis with these 3 stem cell markers may be misleading. Therefore, the description of these data was moved to the supplementary information and extensive discussion on this was avoided. Nevertheless, RNAseq analysis was performed and clustering according to whole transcriptome data, indicated similar trends. These new data were also mentioned in the revised manuscript and are shown as a new Figure 3-Figure Suppl. 2.

Another limitation is the use of cancer cell lines. These cells are already tumorigenic and the influence of pair disruption in this instance is therefore testing whether it can make these cells more or less tumorigenic. This is different than the real-life scenario where it is likely that spouses of deceased partners will not have cancer in the first place, but instead, may develop cancer as a result of the stress and other factors associated with pair disruption. Thus, a more appropriate and complementary model would assess the effects of pair disruption on cancer initiation as opposed to development and progression. For example, pair bonded and disrupted animals could be given a carcinogen to assess the rate and incidence of tumor onset. Or, cell lines that represent a pre-malignant state (ie. don't form tumors in nude mice) could be used to assess the effects on progressing these cells to a fully malignant state.

As discussed above, our findings only explain one aspect of the phenomenon, the effect of widowhood in cancer progression. Whether this also influences cancer initiation, while highly relevant and of particular interest, as the reviewer recognizes cannot be assessed by this model. We are exploring feasible options to address this in our lab, including those suggested by the reviewer and we thank him for that (chemical carcinogenesis and premalignant lesions transplantation).

Reviewer #3:[…] Weaknesses:Although the paper has strengths in principle, the weaknesses of the paper are that these strengths are only reflected in a descriptive manner. Epidemiological evidence already suggests that social interactions and especially bonding between couples influence tumorigenesis. The observation in this paper is interesting and confirmatory of cancer patients but not novel. It would have been informative had the authors utilised the models to perform a more in-depth investigation in providing mechanistic insight on how the bonding disruption affects tumour growth. Some experimental data and experimental details require clarification to allow confident interpretation of the described results. The brief discussion was presented mostly as a summary and statement without sufficient literature support and citation, which should be expanded and enhanced.

The main point of this first study was to address if the well-described, by the epidemiological studies, widowhood effect has biological basis, besides the confounding effects of lifestyle changes and homogamy. We think that our results will now pave the way for the implementation of mechanistic studies addressing this phenomenon. Some mechanistic insights though we believe are provided by the inclusion of RNAseq studies in the revised manuscript.

The manuscript submitted by Naderi et al. entitled "Persistent effects of pair bonding in lung tumorigenesis in monogamous rodents" includes interesting observations that pair-bonding between monogamous rodents (P. californicus) is beneficial to the mice in preventing the implanted tumor growth. However, the current version of the submitted manuscript includes the only limited scope and lack of mechanistic insight. Multiple experimental details were missing that prevent a confident interpretation of the described results.1. In figure 1b, it is not clear when the serum from the bonding (B) group was collected (only one B condition was shown). The appropriate control should include serum from the bonding group at all time points (24h, 1week, and 2 weeks) as the bonding disruption (BD) group for a direct comparison.

We agree with the reviewer’s point about a detailed evaluation of the temporal dynamics of bonding and disruption. In this study we wanted to simulate widowhood and therefore we used a setting of extended bonding, followed by a “small-scale” time course of disruption (1day, 1 week and 2 weeks). Nevertheless, such detailed assessment is in our future plans. In addition, more details about the experimental conditions are now included in the Results, besides the Methods section of the manuscript.

2. There is a mistake in figure 2b. How could it be possible that the 6 mice with the measurable tumor in the virgin (V) group only accounted for 25% of the tumor-bearing sub-group in the pie chart when the total mice number was 8? Based on the results in Figure 2a, the mouse number with a tumor should be 6 and accounted for 75% (6/8) of the total virgin group. Page 5, sentence 99-105 should be better included in the discussion than in the Results section. In addition, the difference is only between 82% (B) and 75% (V), rather than the described 25%, which did not seem to be significant.

We apologize for the confusion in the description of the tumorigenicity results which was due to the fact that the numbers referred to different time points (day 15 and 25, at which several virgins lost their tumors). This point is now better clarified in more detail in the revised manuscript (and the corresponding figure). Per the reviewer’s suggestion the oxytocin-related discussion was transferred from the results to the Discussion section.

3. In figure 2d, how many tumors were transplanted? How many nude mice were used? The information was not included in the methods, nor in the results or figure legends. All of the "N" should be clearly described in the figure legends. Were all tumor cells isolated from all 9 tumor-bearing mice in the B group and 8 tumor-bearing mice from the bonding disruption (BD) group dissected and used for transplantation study? How many different tumor cells from each group (V, B, and BD) were transplanted into nude mice? How many nude mice were transplanted with tumor cells from each tumor-bearing mouse?

We thank the reviewer for pointing these issues that required clarification. All these details are now described in the Results section. In addition (n), besides the results are also described in the figure legend. As now described in the manuscript each Peromyscus-grown tumor was implanted in a single nude mouse.

4. It is not clear how many times the in vivo animal experiments were performed? Did the in vivo data showing in figure 2 represent only a one-time experiment?

Actually, due to the limited availability of deer mice at a given time point, the experiment was performed serially, in 2-3 phases and results were pooled together: As soon as mice became available they were distributed in groups and the experiment was performed. Unfortunately, these are outbred animals at which birth of pups in the colony does not occur simultaneously to allow full synchronization of all experiments. However, as such, we feel that the chances for reflecting a “one-time” observation due to methodological issues, are limited.

5. In figure 3, the signature using only three genes was overly simplified. The selected markers did not provide supporting information on why serum from the different mice groups displayed various sizes in spheroids and tumors because the gene expression of the BD groups was consistently located in between B and V groups, which all showed smaller tumor sizes.6. The clustering description of figure 3 was confusing and the signature was not unique, which was not unexpected because the expression data of these genes did not significantly differ among the different groups. Non-biased analysis using sequencing results with unsupervised clustering likely will provide data that are more informative and mechanistic.

We fully agree with these 2 comments (#5 and #6) and therefore, in the revised manuscript, as also described in our responses to the 1^st^ and 2^nd^ reviewer, the discussion of clustering data, based on CSC markers, were “scaled down” and added in the supplementary information. In addition, as the reviewer suggested, we performed RNA sequencing and the results corroborated our original observations on the clustering based on bonding experiences. Importantly, these analyses provided insights regarding the potential mechanisms, pointing to effects in cell migration and adhesion, and tissue morphogenesis. Furthermore, they indicated that clustering occurs by using whole transcriptome RNAseq data and is more intense between animals (sera) from the bonded and bond-disrupted groups, while virgins have a less “rigid” behavior.

7. The sentence from lines 127-130 is an over-interpretation of the results. The description of transcription could be better included in the discussion and not in the Results section.

We agree. The corresponding description was changed to avoid potential over-interpretation of the results and was included in the Discussion, based primarily on the whole transcriptome data (Discussion, paragraph 5).

8. In figure 4, the description of the serum collection was absent. When was the sera collection time? Were the polygamous mice housed individually after disruption (probably yes) and separated from all-female mice. The figure legend was incomplete and lack important details. The abbreviation should also be described.

We thank the reviewer for pointing this out. We have now included all this missing information in the legend and the Results section.

9. The data in figure 4 regarding the effects on different shapes of spheroids (but without clear biological consequence) did not positively support the hypothesis. Additional mouse species may be needed to identify if the tumor size difference observed in P. californicus after bonding disruption was an exception or a general observation. Alternatively, the authors should limit the claim and re-write the title to reflect the observation in the specific mouse strain.

We agree that the fact that disruption affected differently the spheroids in the two monogamous species was somewhat puzzling. Thus, we did not insist on interpreting the actual effect but rather its occurrence, in the monogamous species only. We note though that it is always possible that this is actually the same effect but recorded at different stages (cell dispersion seen in initially in maniculatus and later, after cells proliferated in californicus as well). It is conceivable that a rigorous time course and dose-response experiment may be able to address all these differences. While these are in our future plans, at this point we feel that may be sufficient to report our observations by emphasizing that an effect was seen. To clarify this we also added in the corresponding description the following:

(Results, 6 lines before the end): “…In addition, it may reflect the same effect (cell dispersion followed by proliferation) but recorded at different stages during the formation of the spheroids.”

In addition, we repeated the experiment and performed statistical analysis on the frequency of its occurrence (n=12 animals for PO and 12 for BW; P=0.0004, chi-sq test; Figure 4-Figure Suppl. 1) which also indicated that P. maniculatus occasionally exhibit the “scattered” phenotype at bonding as well.

Finally, the title was changed, as suggested.

10. In figure 4, the serum collected from the "same animal" at bonding and following the bond disruption did not take into consideration of aging and hormonal effects. Additional groups of animals should have been included to address these concerns and avoid the over-interpretation of the results.

We thank the reviewer for this comment and we have to mention that actually, we believe that hormonal effects of bonding and bond disruption indeed constitute the mechanistic basis for our findings. As regards the age effects, although it remains formally plausible, for animals that are about 1.5 years old (at bonding when serum samples were obtained), serum sampling 1- 2 weeks later appears unlikely to reflect age-related effects. This was also addressed in the B-BD comparison of the results in new Figure 1-Figure Suppl. 1 at which different animals (but sibling pairs) were used.

11. The brief discussion was presented mostly as a summary and statement without sufficient literature support and citation, which should be expanded and enhanced.

We agree with the reviewer that the discussion of our original version of the manuscript was quite descriptive and not very critical and insightful. To some extent this was due to the fact that we wanted to refrain from overestimating some findings and engage in speculations. Nevertheless, in this revised version of our paper the Discussion section was significantly changed and a more critical approach was applied.